# Haploidentical versus Double-Cord Blood Stem Cells as a Second Transplantation for Relapsed Acute Myeloid Leukemia

**DOI:** 10.3390/cancers15020454

**Published:** 2023-01-10

**Authors:** Jong-Hyuk Lee, Byung-Sik Cho, Daehun Kwag, Gi-June Min, Sung-Soo Park, Silvia Park, Jae-Ho Yoon, Sung-Eun Lee, Ki-Seong Eom, Yoo-Jin Kim, Seok Lee, Chang-Ki Min, Seok-Goo Cho, Jong-Wook Lee, Hee-Je Kim

**Affiliations:** 1Department of Hematology, Catholic Hematology Hospital, Seoul St. Mary’s Hospital, College of Medicine, The Catholic University of Korea, Seoul 06591, Republic of Korea; 2Leukemia Research Institute, College of Medicine, The Catholic University of Korea, Seoul 06591, Republic of Korea

**Keywords:** acute myeloid leukemia, haploidentical donor, double-cord blood, second stem cell transplantation

## Abstract

**Simple Summary:**

Second stem cell transplantation (SCT2) may provide long-term remission for patients with relapsed acute myeloid leukemia (AML) after the first transplantation (SCT1). There are increasing demands for alternative donors, namely haploidential and umbilical cord blood, even in SCT2. In this single-center retrospective analysis for AML patients who relapsed after SCT1, we compared SCT2 outcomes with haploidentical donors (HIT) or double-cord blood (dCBT). In our study, HIT had superior transplant outcomes to dCBT as SCT2 in AML, due to the high non-relapse mortality in the dCBT group, which resulted in poorer survival. In a subgroup analysis, pre-transplant *WT1*-MRD positivity was associated with higher relapse rates and worse outcomes.

**Abstract:**

There are limited data on second stem cell transplantation (SCT2) outcomes with alternative donors for relapsed AML after the first stem cell transplantation (SCT1). We analyzed the outcomes of 52 adult AML patients who received SCT2 from haploidentical donors (HIT, *N* = 32) and double-cord blood (dCBT, *N* = 20) between 2008 and 2021. The HIT group received T-cell-replete peripheral blood stem cells after reduced-toxicity conditioning with anti-thymocyte globulin (ATG), while the dCBT group received myeloablative conditioning. For a median follow-up of 64.9 months, the HIT group, compared to the dCBT group, had earlier engraftment, superior 2-year overall survival (OS), disease-free survival (DFS), and non-relapse mortality (NRM) with similar relapse. Multivariate analysis demonstrated that HIT was significantly associated with better OS, DFS, and lower NRM than dCBT. Both longer remission duration after SCT1 and complete remission at SCT2 were significantly associated with a lower relapse rate. In addition, bone marrow *WT1* measurable residual disease (MRD) positivity was significantly associated with inferior OS and higher relapse. This study suggests that T-cell-replete HIT with ATG-based GVHD prophylaxis may be preferred over dCBT as SCT2 for relapsed AML and that *WT1*-MRD negativity may be warranted for better SCT2 outcomes.

## 1. Introduction

Relapse after stem cell transplantation (SCT) is a major cause of treatment failure in patients with acute myeloid leukemia (AML) and generally leads to poor survival outcomes [1,2]. Conventional chemotherapy and novel agents for post-SCT relapse in AML patients exhibit limited efficacy, especially in terms of achieving a durable response [1,3,4,5]. A recent observational study from the European Society for Blood and Marrow Transplantation (EBMT) compared a second allogeneic SCT with donor lymphocyte infusion (DLI) in relapsed AML after allograft. While both treatment modalities demonstrated similar overall survival (OS), DLI recipients had a significantly lower rate of remission compared to those who underwent a second SCT (SCT2) [3]. Most studies on SCT2 for hematologic malignancies repeatedly identified short remission duration after first SCT (SCT1) and active disease status before SCT2 as prognostic predictors of inferior outcomes, while there were contradictory results on the prognostic role of age, conditioning intensity at SCT1, prior graft-versus-host disease (GVHD) after SCT1, and transplant from a human leukocyte antigen (HLA)-mismatch donor [1,2,3,6,7,8,9,10,11,12,13,14,15,16,17,18].

Transplantation with HLA-mismatched alternative donors, such as haploidentical (HIT) and umbilical cord blood (CBT), has been increasingly used for patients without a suitable HLA-matched donor. Historically, HLA-mismatched transplantations were associated with higher incidences of GVHD, non-relapse mortality (NRM), and inferior survival. However, T-cell-replete strategies using anti-thymocyte globulin (ATG) and/or post-transplant cyclophosphamide (PTCy) as GVHD prophylaxis in HIT seem to overcome these disadvantages, leading to non-inferior outcomes as in the setting of SCT1 [19,20]. Umbilical cord blood is readily available and used across HLA barriers, with one of the main reasons being the limited number of T-cells. Double-cord blood unit infusions are frequently applied to overcome the small number of stem cells in a single unit [21,22,23,24,25]. However, there are limited data on the outcomes of SCT2 using these alternative donors in relapsed AML. Several retrospective studies for T-cell-replete HIT using various conditioning regimens with PTCy-based GVHD prophylaxis [14,16,26] and no data for CBT have been reported in this setting. There have yet to be any reports comparing HIT with CBT as SCT2. We developed T-cell-replete HIT using a unique conditioning regimen that is based on fludarabine, busulfan, and an intermediate dose of total body irradiation (TBI) [27,28]. The reduction in the TBI dose from a classical TBI-based myeloablative conditioning (MAC) can reduce nonhematologic toxicity; many authors frequently describe such conditioning intensity as reduced toxicity conditioning (RTC) [29,30,31,32,33,34,35]. Using the RTC regimen, along with ATG-based GVHD prophylaxis, we reported comparable transplant outcomes to matched unrelated donors in a prospective, biologically randomized study [19]. In this study, we compared the outcomes of T-cell-replete HIT using ATG-based GVHD prophylaxis with CBT using double units (dCBT) in relapsed AML patients after SCT1.

## 2. Results

### 2.1. Patient Characteristics

A total of 52 patients were included in this study. Patient-, disease-, and treatment-related characteristics at the time of diagnosis, SCT1, and SCT2 are described in Table 1. The median age at SCT2 was 47 years (range 23–67 years) and 53.8% of the patients were male (*N* = 28). The HIT (*N* = 32) and dCBT (*N* = 20) groups had no difference in the median age of patients (46.5 years vs. 49 years; *p* = 0.785). The patients in the HIT group were more likely to have received autologous transplant and myeloablative conditioning at SCT1 compared to patients in the dCBT group (41% vs. 5%, *p* < 0.001; 78% vs. 25%, *p* < 0.001, respectively). However, there was no difference between the groups with respect to cytogenetic risk, FLT3-ITD mutations, duration of remission after SCT1, intensity of salvage therapy, disease remission status before SCT2, and time from relapse to SCT2. The total infused nucleated cells and CD34+ cells were significantly greater in the HIT group compared to the dCBT group at SCT2 (*p* < 0.001 for both types).

### 2.2. Engraftment, GVHD, and Other Complications

The dCBT group exhibited a significantly delayed recovery of neutrophil and platelet counts compared to the HIT group (Figure 1A,B). The median (range) time to neutrophil engraftment were 12 days (6–23) for the HIT group and 28.5 days (16–75) for the dCBT group (*p* < 0.001). The median (range) days to platelet engraftment were 13 days (8–39 days) for the HIT group and 52 days (18–167 days) for the dCBT group (*p* < 0.001). The cumulative incidences of acute GVHD and grade 2–4 acute GVHD at 6 months were similar in each group (HIT vs. dCBT; acute GVHD, 41% vs. 45%, *p* =.866, Figure 1C; grade 2–4 acute GVHD, 38% vs. 33%, *p* = 0.426). The HIT group exhibited more frequent all grade chronic GVHD and moderate or severe chronic GVHD at 2 years compared to the dCBT group but without statistical significance (chronic GVHD, 46% vs. 15%, *p* = 0.057, Figure 1D; moderate or severe chronic GVHD, 29% vs. 6%, *p* = 0.123). There was no significant difference in CMV-related events (DNAemia, disease, or receiving anti-CMV treatment), thrombotic microangiopathy, sinusoidal obstruction syndrome, or herpes zoster infection between the two groups (Appendix A). However, hemorrhagic cystitis occurred more frequently in the HIT group compared to the dCBT group (31% vs. 5%, *p* = 0.035).

### 2.3. Survival Outcomes between HIT and dCBT

The median (range) follow-up period from the time of SCT2 for all the patients was 64.9 months (12.6–151.6 months) for survivors. The 2-year cumulative incidence of NRM was lower in the HIT group compared to the dCBT group (22% vs. 50%, *p* = 0.014; Figure 1E), while there was no difference in the 2-year CIR (34% vs. 31%, *p* = 0.747, Figure 1F). The 2-year OS and 2-year DFS were significantly better for the HIT group than the dCBT group (43% vs. 30%, *p* = 0.018, Figure 1G; 44% vs. 19%, *p* = 0.023, Figure 1H, respectively). The early NRM (<100 days after transplant) was higher without statistical significance in dCBT than HIT (25% vs. 9%, *p* = 0.235). Infection was the most common cause of NRM in both the HIT and dCBT groups (16% vs. 35%, *p* = 0.175) with a higher tendency in dCBT, but no significant differences in NRM causes between the two groups were observed (Appendix A). All the patients who relapsed died (*N* = 11, 34.3% of the HIT group; *N* = 6, 30.0% of the dCBT group), except for one patient in the HIT group, who is currently alive at more than 2 years after the third allogeneic transplant with double-cord blood.

### 2.4. Factors Affecting Survival Outcomes

Forest plots of the univariate analysis to identify risk factors associated with SCT2 outcomes (OS, DFS, NRM, and CIR) were presented in Appendix A. HIT, compared to dCBT, was associated with lower NRM (hazard ratio [HR] 0.34, 95% confidence interval [CI] 0.14–0.84, *p* = 0.019), resulting in superior OS (HR 0.46, 95% CI 0.23–0.89, *p* = 0.021) and DFS (HR 0.47, 95% CI 0.24–0.92, *p* = 0.027). Older age at SCT2 was associated with worse NRM (HR 1.04, 95% CI 1.01–1.07, *p* = 0.010) while disease remission at SCT2 was associated with less CIR (HR 0.25, 95% CI 0.10–0.63, *p* = 0.003). For patients in remission before SCT2 with available *WT1*-MRD data, *WT1* greater than 250 copies/10^4^
*ABL* at SCT2 resulted in worse OS, DFS, and CIR (HR 2.96, 95% CI 1.23–7.12, *p* = 0.016; HR 3.47, 95% CI 1.44–8.38, *p* = 0.006; HR 4.87, 95% CI 1.50–15.8, *p* = 0.008, respectively). Prior acute or chronic GVHD at SCT1 was not associated with NRM (acute GVHD grade ≥ 2: HR 0.76, 95% CI 0.17–3.48, *p* = 0.720; moderate or severe chronic GVHD: HR 0.91, 95% CI 0.29–2.88, *p* = 0.871).

On multivariate analysis (Figure 2), HIT was significantly associated with lower NRM (HR 0.29, 95% CI 0.12–0.74, *p* = 0.009) and superior OS (HR 0.43, 95% CI 0.22–0.84, *p* = 0.014) and DFS (HR 0.49, 95% CI 0.25–0.95, *p* = 0.035), compared to dCBT. Older age was associated with worse NRM and OS (in decades, NRM, HR 1.73, 95% CI 1.22–2.46, *p* = 0.002; OS, HR 1.01, 95% CI 1.01–1.80, *p* = 0.044). Disease in remission at SCT2 (HR 0.23, 95% CI 0.10–0.56, *p* = 0.001) as well as longer than 9 months of remission after SCT1 (HR 0.41, 95% CI 0.17–0.99, *p* = 0.047) resulted in less CIR. On a separate univariate (Appendix A) and multivariate analysis (Figure 3) with 38 patients in remission with available WT1-MRD status before SCT2, the WT1-MRD positivity (>250 copies/104 ABL) was significantly associated with worse OS (HR 4.56, 95% CI 1.78–11.7, *p* = 0.002), DFS (HR 6.92, 95% CI 2.48–19.3, *p* < 0.001), and CIR (HR 5.82, 95% CI 1.88–18.1, *p* = 0.002).

### 2.5. Subgroup Analysis with Patients Who Underwent Allogeneic SCT1

Given that the HIT group received more autologous transplantation for SCT1 compared to the dCBT group, we performed another subgroup analysis with patients who underwent allogeneic SCT1 (HIT group, *N* = 19; dCBT group, *N* = 19, Appendix A). Among patients with allogeneic SCT1, no significant difference was observed in GVHD occurrence after SCT1 (HIT vs. dCBT; acute GVHD grade ≥ 2: 16% vs. 21%, *p* = 1.000; moderate or severe chronic GVHD: 16% vs. 32%, *p* = 0.447). One patient in HIT group experienced a recurrence of acute skin GVHD after salvage therapy and eventually died due to disease relapse after SCT2. The subgroup analysis (Appendix A) showed similar trends of superior 2-year OS (41% vs. 26%, *p* = 0.074) and DFS (42% vs. 14%, *p* = 0.111) in the HIT group compared to the dCBT group as in the whole group but without statistical significance. We observed lower rates of NRM (21% vs. 52%, *p* = 0.068) in the HIT group than in the dCBT group with similar CIR (37% vs. 33%, *p* = 0.745). There were no significant differences in acute GVHD and chronic GVHD in both groups. Prior GVHD did not affect the aforementioned survival outcomes (Appendix A).

In addition, a subgroup analysis (Appendix A) determining whether autologous or allogeneic SCT1 would have an effect on outcomes after HIT as SCT2 showed no significant difference in 2-year OS (auto vs. allo, 46% vs. 41%, *p* = 0.480), DFS (46% vs. 42%, *p* = 0.293), NRM (23% vs. 21%, *p* = 0.737), and CIR (31% vs. 37%, *p* = 0.577), or in acute and chronic GVHD.

## 3. Discussion

This retrospective single-center cohort study investigated the clinical outcomes of SCT2 with HIT and dCBT for AML patients who relapsed after SCT1. OS, the primary endpoint, exhibited better results in HIT compared to dCBT, primarily due to the higher NRM in dCBT. Neutrophil/platelet engraftment and NRM favored the HIT group, while CIR and acute/chronic GVHD did not differ between the two groups. Advanced disease status at SCT2 and a shorter time to relapse after SCT1 (less than 9 months) were significantly associated with a higher CIR. In addition, subgroup analysis revealed that a *WT1*-MRD status was predictive for post-transplant relapse and subsequent survival outcomes in this setting.

This is the first report, to the best of our knowledge, on T-cell-replete HIT as SCT2 using ATG-based GVHD prophylaxis in AML. Our HIT protocol was also characterized using a unique reduced-toxicity regimen including fractionated TBI 800 cGy with T-cell-replete PBSC and validated in a prospective study demonstrating the equivalence in survival outcomes with matched unrelated SCT for AML in the setting of SCT1 [19]. The advantages of our HIT protocol in SCT1, such as perfect engraftment with rapid neutrophil and platelet recovery, were again observed in this study of SCT2 for relapsed AML after SCT1. Several retrospective studies have described outcomes of HIT as SCT2 in relapsed hematologic malignancies (Appendix A) [14,16,26]. They commonly used PTCy-based GVHD prophylaxis but with differences in conditioning intensity and stem cell sources, contrasting our study of relapsed AML using a uniform protocol consisting of ATG-based GVHD prophylaxis with T-cell-replete PBSC for HIT. A direct comparison may be limited due to the wide range of clinical factors, such as baseline patient and disease characteristics, the proportion of AML patients in the study population, time from the first transplant to relapse, salvage therapies before SCT2, remission status before SCT2, conditioning intensity, and proportion of PBSC as the stem cell source. Nonetheless, estimated OS after at least one year ranged from 29% to 45% in those studies, which is comparable to our 2 year-OS of 43%. CIR, NRM, and acute GVHD were similar as well, while the incidence of chronic GVHD in our cohort was higher than in previous studies. The current study first demonstrates that HIT using T-cell-replete PBSC and ATG-based GVHD prophylaxis has comparable outcomes for SCT2 in AML patients compared to those with HIT using PTCy-based GVHD prophylaxis. Given that the EBMT registry data reported better outcomes in leukemia-free survival, GVHD-free and relapse-free survival, GVHD, and NRM for PTCy-based regimens compared to ATG-based GVHD prophylaxis in AML for SCT1 [38], the superiority of PTCy over ATG is not yet conclusive. Prospective randomized trials comparing PTCy- and ATG-based GVHD prophylaxis for T-cell-replete HIT are warranted in the setting of SCT2 as well as SCT1. 

This is also the first study to describe the outcomes of dCBT as SCT2 in AML, which was compared with HIT as SCT2. Our dCBT protocol has been utilized for acute lymphoblastic leukemia (ALL) in first CR with comparable survival outcomes with transplants from HLA-matched siblings or unrelated donors [25,39]. Lower incidences of chronic GVHD and CIR, but higher NRM, were characterized in dCBT as SCT1 for ALL in CR1 compared to other donor types. Older age over 40 was associated with higher early death (less than 100 days from transplant) and NRM. Many studies have been conducted to investigate the role of reduced-intensity conditioning (RIC) in cord-blood transplantation to reduce NRM and improve survival [40,41,42]. However, large registry data in AML showed that RIC did not significantly reduce NRM [37,38] but increased the risk of relapse [38] compared to MAC. Several comparative data sets for HIT and CBT in SCT1 for hematologic malignancies have been published [39,43,44,45,46,47,48,49]. Japanese [46] and European [49] registry data reported that CBT and HIT exhibited similar survival outcomes in adult AML patients. On the other hand, a recent BMT CTN 1101 trial, a randomized multicenter phase 3 trial of acute leukemia and lymphoma in remission comparing dCBT and HIT from BM using different RIC regimens in both donor types, showed that NRM and OS favored HIT over dCBT with statistical significance [44]. Indeed, CBT has concerns about a higher NRM compared to other donor types, primarily due to delayed immune reconstitution and increased risk for graft failure [43,44,45,47]. In line with the randomized phase 3 trial in SCT1 [44], our study demonstrates that delayed engraftment and higher NRM in dCBT lead to inferior OS compared to HIT in the setting of SCT2, even after age adjustment in multivariate analysis. The dCBT group had higher early NRM than the HIT group. However, previously observed stronger graft-versus-leukemia effects of dCBT versus other donor types in SCT1 for ALL [39] were not observed in AML when compared to HIT for SCT2. These findings suggest that T-cell-replete HIT with ATG-based GVHD prophylaxis would be the preferred approach in the setting of SCT2. Additional randomized controlled trials must be conducted to verify our results. Moreover, HIT has a benefit over dCBT in providing platforms for prophylactic or pre-emptive interventions using donor-originated cells, such as DLI [50], NK cells [51], or antigen-specific T cells [52]. On the other hand, given the higher CIR than NRM in HIT as SCT2 in this study, antibody or cell-based immunotherapy [50,51,52] and post-transplant maintenance therapy [53] under clinical trials must be considered to prevent relapse in SCT2. 

Our data revealed that the disease status before SCT2 and time from SCT1 to relapse were significant prognostic factors in the setting of SCT2. Our group previously reported that later relapse (greater than 9 months) from the first transplant, younger age, and remission at SCT2 were associated with superior survival in AML patients who relapsed after SCT1 [11]. In line with these findings, the current study using alternative donors showed that longer times to relapse after SCT1 and disease in remission at SCT2 were significantly associated with lower CIR. Given the conflicting results about the impact of time from the first transplant to relapse on outcomes of HIT for SCT2 [14,16,17,26], our data support the impact of time from SCT1 to relapse. Notably, subgroup analysis with patients in remission at SCT2 revealed that pre-transplant *WT1*-MRD status was a significant factor for post-transplant relapse and poorer survival. Our group reported the prognostic role of *WT1*-MRD assessment in SCT1 for AML [19,54,55], which was consistent in this study with SCT2. These findings suggest the importance of quality in response to salvage treatment at relapse after SCT1 for the success of SCT2. Various emerging treatments, such as targeted therapy and immunotherapy [56], would be applicable to induce a deeper remission through various combinations with existing treatments. In addition, these emerging treatments would be applicable after SCT2 as a pre-emptive therapy based on MRD monitoring or prophylactic maintenance therapy. In this regard, relapsed AML patients after SCT1 should be enrolled in clinical trials with novel therapeutic strategies if possible. 

This study has several limitations, including its retrospective nature, small sample size, imbalance in conditioning intensity, long treatment period, and limited genomic profiling. The donor selection process may have introduced bias into the results, as cord-blood transplants were provided when neither an HLA-matched donor nor a haploidentical donor was available. Given the limited reports of SCT2 using alternative donors and the lack of prospective data, our findings showed that T-cell-replete HIT with ATG-based GVHD prophylaxis had better outcomes than dCBT in the setting of SCT2. Both the HIT and dCBT groups were well balanced except for transplant type at SCT1 and conditioning intensity at SCT2. Autologous transplantation was more frequently performed for SCT1 in the HIT group compared to the dCBT group. However, there were no significant differences in the survival outcomes of SCT2 between patients with autologous and allogeneic SCT1 in the HIT group. Furthermore, a subgroup analysis including patients who underwent allogeneic SCT1 in both groups revealed a similar trend of superiority of the HIT group over the dCBT group in survival outcomes. Based on the differences in transplant protocols, including conditioning intensity, the outcomes between HIT and dCBT should be interpreted as results not from different donor types but from the transplant strategy to utilize stem cells from different donors. Given the aforementioned limitations, the findings in this study must be confirmed in a large prospective study.

## 4. Conclusions

Our findings show that T-cell-replete HIT with ATG-based GVHD prophylaxis would be the preferred approach in SCT2 compared to dCBT. Given the importance of a higher quality of response to salvage treatment at relapse for the success of SCT2 in this study, various emerging treatments, such as novel targeted therapy and immunotherapy, must be actively incorporated into the standard treatment for relapsed AML patients after SCT1. In addition, adjuvant treatment with prophylactic or pre-emptive strategies after transplant to prevent relapse may improve outcomes after SCT2.

## 5. Materials and Methods

### 5.1. Patients

We retrospectively analyzed data from consecutive patients with AML who underwent SCT2 from either haploidentical donors or double-cord bloods for relapse after SCT1 between 1 January 2008 and 31 December 2021. The patients were first assigned to HIT in the absence of HLA-matched sibling or unrelated donors and secondarily to dCBT in the absence of available haploidentical donors. Cord-blood units were selected as previously described [39]. The patients have received either autologous or allogeneic SCT1 and a different donor from SCT1 was selected for SCT2 according to a previously described strategy from our institution [11]. This study was approved by the Institutional Review Board of Seoul St. Mary’s Hospital in accordance with the principles of the Declaration of Helsinki. The survival data had been updated as of 30 June 2022.

### 5.2. Transplantation Procedures

The conditioning regimens and GVHD prophylaxis for HIT and dCBT were described in Appendix A. The HIT group received RTC with fractionated TBI (8.0 Gy), fludarabine (150 mg/m^2^), intravenous busulfan (6.4 mg/kg), and ATG (5.0 mg/kg; Sanofi-Genzyme, Lyon, France) for in vivo T-cell depletion and tacrolimus with methotrexate for GVHD prophylaxis. The dCBT group received myeloablative conditioning (MAC) with fractionated TBI (12.0 Gy), fludarabine (150 mg/m^2^), and cytarabine (9.0 g/m^2^) and tacrolimus and mycophenolate mofetil for GVHD prophylaxis. None of the patients received ex vivo T-cell depletion or PTCy. For the HIT group, we used peripheral blood as the stem cell source (PBSC). For all the patients undergoing dCBT, double-cord blood units were infused. The details for cord-blood selection, transplant procedures, and *WT1* quantification in bone marrow (BM) for measurable residual disease (MRD) assessments were previously described [11,19,25,39,54]. 

### 5.3. Definitions

Remission, representing complete hematologic remission (CR), was defined by the International Working Group response criteria for AML [57]. Relapse was defined as the reappearance of leukemic blasts in the peripheral blood or ≥5% infiltration of a representative BM smear. Neutrophil engraftment was defined as an absolute neutrophil count >0.5 × 10^9^/L in the first three consecutive days and platelet engraftment was defined as a platelet count >20 × 10^9^/L in the first of five consecutive days without transfusions. OS was defined as the time from the second transplantation to death from any cause. Disease-free survival (DFS) was defined as survival without evidence of relapse or progression. Relapse was defined as the recurrence of leukemia, regardless of site. NRM was defined as death without evidence of relapse.

### 5.4. Statistical Analysis

Patient-, disease-, and treatment-related variables of the groups (HIT vs. dCBT) were compared using χ^2^ or Fisher exact test for categorical variables and the Mann–Whitney test for continuous variables. Baseline characteristics were summarized using the median and range for continuous variables and numbers and frequencies for categorical variables. The primary end point was OS. Secondary end points included DFS, cumulative incidence of relapse (CIR), NRM, acute GVHD, chronic GVHD, and neutrophil/platelet engraftment. Death for CIR, relapse for NRM, and both death and relapse for GVHD/engraftment were treated as competing events for cumulative incidence estimation. The survival outcomes were calculated using the Kaplan–Meier method, while the cumulative incidence function was used to estimate the CIR, NRM, acute GVHD, and chronic GVHD. Univariable analysis was performed using the logrank test for OS and DFS and the Gray test for CIR and NRM. The variables with a *p*-value less than 0.1 were included in multivariable analysis. The multivariable analyses were performed using the Cox proportional hazard regression model for OS and DFS, while the Fine-Gray method was used for relapse and NRM. All the statistical analyses were performed using R 4.1.2 (R Foundation for Statistical Computing, Vienna, Austria; http://www.r-project.org/, accessed on 30 July 2022) and EZR 1.55 [58]. The statistical significance was set at a *p* value less than 0.05.

## Figures and Tables

**Figure 1 cancers-15-00454-f001:**
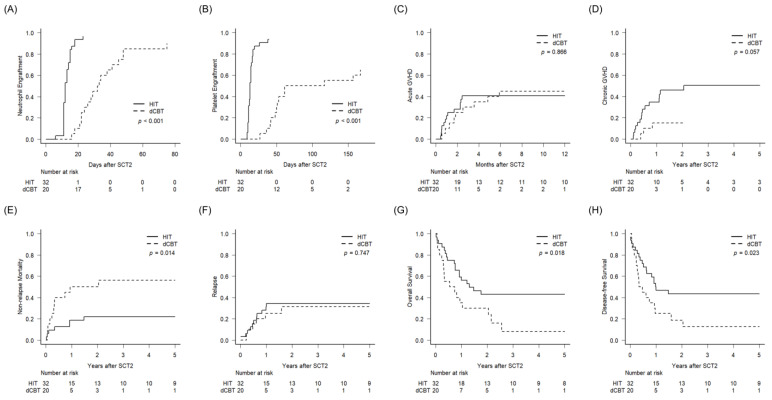
Outcomes after haploidentical and double-cord blood transplantations as the second transplantation. The cumulative incidences of (**A**) neutrophil recovery, (**B**) platelet recovery, (**C**) acute GVHD, (**D**) chronic GVHD, (**E**) non-relapse mortality, (**F**) relapse, and probabilities of (**G**) overall survival and (**H**) disease-free survival.

**Figure 2 cancers-15-00454-f002:**
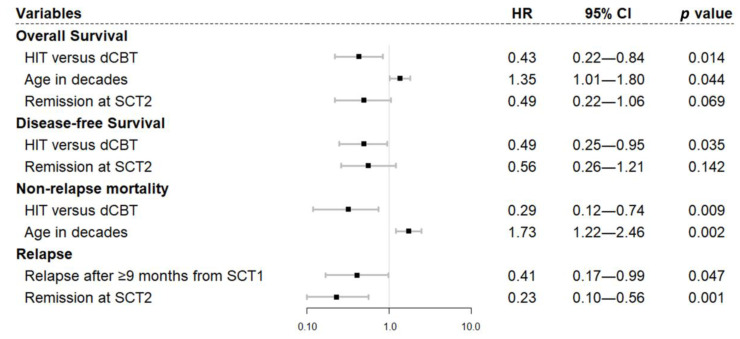
Multivariate analysis of outcomes after second transplantation.

**Figure 3 cancers-15-00454-f003:**
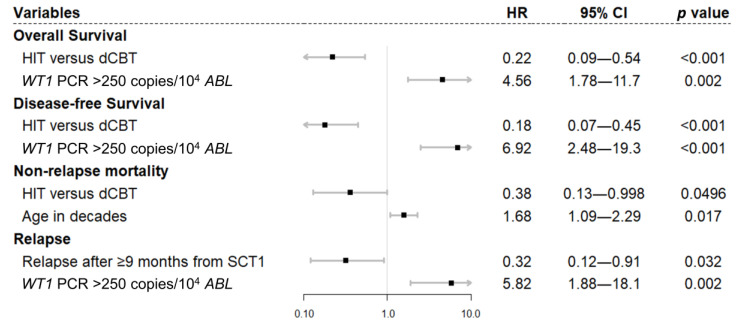
Multivariate analysis of outcomes after second transplantation for patients with available pre-transplant *WT1*-PCR (*N* = 38).

**Table 1 cancers-15-00454-t001:** Patient characteristics.

Characteristics		HIT (*N* = 32), *N* (%)	dCBT (*N* = 20), *N* (%)	*p* Value
AML MRC/treatment-related		5 (16)/1 (3)	5 (25)/0 (0)	0.480
Age at SCT2, median (range)		46.5 (23–67)	49 (25–63)	0.785
Male		15 (47)	13 (65)	0.258
Hyperleukocytosis (≥50 × 10^9^/L) at diagnosis Unavailable CBC at diagnosis		4 (16) ^a^ 7	3 (17) ^a^ 5	1.000
Cytogenetics ^b^	Favorable Intermediate Adverse	4 (13) 25 (78) 3 (9)	3 (15) 10 (50) 7 (35)	0.053
FLT3-ITD	Positive Negative Not available	4 (13) 20 (63) 8 (25)	3 (15) 14 (70) 3 (15)	0.697
Donor at SCT1	Autologous Matched-related Matched-unrelated Haploidentical Cord blood	13 (41) 11 (34) 7 (22) 0 (0) 1 (3)	1 (5) 4 (20) 2 (10) 13 (65) 0 (0)	<0.001
SCT1 Intensity	MAC RTC ^c^ RIC	25 (78) 3 (9) 4 (13)	5 (25) 12 (60) 3 (15)	<0.001
Remission duration after SCT1 ≥9 months		24 (75)	14 (70)	0.754
Salvage therapy	*Intensive* FLANG ^d^ FLAG-Ida ^e^ MEC ^f^ IA ^g^ *Non-intensive* Venetoclax-based Sorafenib-based With DLI	30 (94) 22 (69) 1 (3) 4 (13) 3 (9) 2 (6) 1 (3) 1 (3) 2 (6)	18 (90) 14 (70) 0 (0) 2 (10) 1 (5) 2 (10) 2 (10) 0 (0) 0 (0)	0.6340.517
Complete remission before SCT2		25 (78)	17 (85)	0.722
*WT1* PCR (/10^4^ ABL) before SCT2 CR, ≥250 (MRD-positive) CR, <250 (MRD-negative) CR, but data unavailable Non-CR		5 (16) 18 (56) 2 (6) 7 (22)	3 (15) 13 (65) 1 (5) 3 (15)	0.960
HCT-CI	0 1–2 ≥3	5 (16) 12 (38) 15 (47)	5 (25) 3 (15) 12 (60)	0.207
Donor sex	Match Partial mismatch Full mismatch	16 (50) - 16 (50)	3 (15) 12 (60) 5 (25)	
Donor HLA match (of eight loci, the lower value in dCBT)	4 5 6 ≥7	11 (34) 16 (50) 5 (16) 0 (0)	3 (15) 7 (35) 6 (30) 4 (20)	<0.001
Total nucleated cells, median (range), 10^8^/kg		20.5 (8.7–40.3)	0.39 (0.18–0.71)	<0.001
CD34, median (range), 10^6^/kg		7.44 (1.77–28.96)	0.07 (0.03–0.32)	<0.001
Time between SCT1 and SCT2, median months (range)		17.2 (6.6–142.0)	15.7 (6.6–82.0)	0.333
Time between relapse and SCT2, median months (range)		3.7 (2.8–7.9)	3.7 (2.8–7.1)	0.631

Abbreviations: CBC: complete blood count; CR: complete remission; dCBT: double-cord blood transplantation; DLI: donor lymphocyte infusion; FLT3-ITD: fms-related tyrosine kinase 3-internal tandem duplication; GVHD: graft-versus-host disease; HCT-CI: hemotopoietic cell transplantation-specific comorbidity index; HIT: haploidentical donor transplant; HLA: human leukocyte antigen; MAC: myeloablative conditioning; MRC: myelodysplasia-related changes; PCR: polymerase chain reaction; RIC: reduced-intensity conditioning; RTC: reduced-toxicity conditioning; SCT1: first stem cell transplantation; SCT2: second stem cell transplantation; TNC: total nucleated cells. ^a^ Percentage was calculated as (number of patients with hyperleukocytosis at diagnosis)/(number of patients with available CBC at diagnosis) × 100 (%). ^b^ Cytogenetic classification was based on the International Working Group (IWG) response criteria for acute myeloid leukemia, published in 2003 [36]. ^c^ Reduced toxicity conditioning refers to fractionated TBI (8.0 Gy), fludarabine (150 mg/m^2^), and intravenous busulfan (6.4 mg/kg) [19]. ^d^ FLANG regimen consists of fludarabine 30 mg/m^2^/day, cytarabine 1 g/m^2^/day, mitoxantrone 10 mg/m^2^/day, and granulocyte-colony stimulating factor 300 μg/day for 5 days [37]. ^e^ FLAG-Ida regimen consists of fludarabine 30 mg/m^2^/day, cytarabine 1 g/m^2^/day, idarubicin 12 mg/m^2^/day and granulocyte-colony stimulating factor 300 μg/day for 5 days. ^f^ MEC regimen consists of cytarabine 2 g/m^2^/day, mitoxantrone 10 mg/m^2^/day for 4 days followed by etoposide 100 mg/m^2^/day for 3 days. ^g^ IA regimen refers to idarubicin 12 mg/m^2^/day for 3 days and cytarabine 100 mg/m^2^/day for 7 days.

## Data Availability

The data that support the findings of this study are available from the corresponding author upon reasonable request. The data are not publicly available due to ethical considerations.

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
