# Peer review of "Haploidentical versus Double-Cord Blood Stem Cells as a Second Transplantation for Relapsed Acute Myeloid Leukemia"

_cancers, 2023, doi:10.3390/cancers15020454_

Round 1

Reviewer 1 Report

In the manuscript, Lee and colleagues study a cohort of patients receiving a second haematopoietic stem cell transplant (HSCT) for relapsed acute myeloid leukaemia (AML). As is correctly pointed out, there is increasing interest in alternative donor transplants even for situations requiring a second transplant. The authors find that outcomes among the 32 patients who received a haploidentical transplant were better than those among the 20 patients who received a double umbilical cord transplant, largely because of higher non-relapse mortality among the latter.

Main comment:

As my main comment and criticism, this manuscript reports on a limited number of patients. Together with the fact that the patient characteristics (e.g. the number of patients who received an autologous versus allogeneic HSCT as first transplant; number of patients who receive myeloablative conditioning with the first transplant) as well as transplant characteristics (in particular, intensity of conditioning) differ significantly among the 2 patient cohorts. It therefore appears impossible to draw any conclusion as to whether the observed differences in outcomes between haploidentical and double umbilical cord transplants are due to patient or transplant characteristics differences or the difference in stem cell source. For example, while a subset analysis in this manuscript found no difference (an analysis again limited by small sample size), I would have expected outcomes with a second allograft to be worse than outcomes with a first allograft after prior autologous HSCT. The subset analysis of the patients who received a prior allogeneic HSCT is therefore central (the authors could even consider to limited their manuscript on this subset of patients as the larger analysis does not appear that helpful) – but is unfortunately limited by the very small number of patients (19 in each group).

Minor comment:

1. Please explain the difference between RIC and RTC (table 1).

2. Given that haploidentical HSCTs are nowadays mostly done with post-transplant cyclophosphamide rather than ATG, and outcomes appear better (reference #31), the authors may want to expand the discussion section and explain why they are still using their ATG-based transplant approach.

3. Since WT1 is no longer used for MRD monitoring in many places, I wonder whether there are other measures of MRD (e.g. multiparameter flow cytometry) that were done and could be included?

Reviewer 2 Report

This is a well written manuscript. The described study is certainly limited by the rarity of second transplant as well as its single-center, retrospective nature. In general, the authors do not "over-claim" conclusions for such a small sample and also do a nice job of incorporating discussion about post-transplant cyclophosphamide for haploidentical transplant despite their use of T-cell replete ATG-based approach. 

My one concern is that selection bias is not adequately addressed in the discussion or conclusions. The difference in first autologous transplant is addressed through subgroup analysis, but in reviewing the methods, it seems that patients are offered haploidentical transplant by default and only proceed with cord blood transplant if no haploidentical donor option is available (or was already utilized for first transplant). It is clear that the HIT and dCBT groups are not equivalent in their pre-treatment, but this institutional approach should be addressed throughout the manuscript and not just in the methods. It seems unlikely that there would be a way to measure whether patients were not offered second transplant if they did not have a haploidentical donor given the retrospective nature, but would suggest commenting on this possibility and that this would need to be addressed in a prospective study. BMT CTN 1101 was a good effort to measure the true difference between HIT and dCBT because participants were required to have both haplo and cord blood options and could be randomized.

Reviewer 3 Report

Jong Hyuk Lee et al. investigated the clinical outcomes of second stem cell transplantation with haploidentical donors and double cord blood for AML patients who relapsed after first stem cell transplantation, found that disease status before second stem cell transplantation and time from first stem cell transplantation to relapse were significant prognostic factors in the setting of second stem cell transplantation, showed that T-cell replete haploidentical donors with ATG-based GVHD prophylaxis would be the preferred approach in second stem cell transplantation compared to double cord blood. The conclusion provided some useful information and may benefit our readers in the relevant fields, the reviewer only has minor issues listed below:

1. Related to Figure 1, how about other CBC index? like WBC, RBC etc.

2. Figure legends usually locate under related figures.

Round 2
